# A TCP Acceleration Algorithm for Aerospace-Ground Service Networks

**DOI:** 10.3390/s22239187

**Published:** 2022-11-26

**Authors:** Canyou Liu, Jimin Zhao, Feilong Mao, Shuang Chen, Na Fu, Xin Wang, Yani Cao

**Affiliations:** 1State Key Laboratory of Astronautic Dynamics, Xi’an Satellite Control Center, Xi’an 710043, China; 2Space Star Technology Co., Ltd., Beijing 100086, China; 3Department of Electronic and Optical Engineering, Space Engineering University, Beijing 101416, China

**Keywords:** network delay, packet loss rate, aerospace-ground service network, BoostTCP acceleration algorithm, bottleneck bandwidth and round-trip propagation time congestion control algorithm, cubic congestion control algorithm

## Abstract

The transmission of satellite payload data is critical for services provided by aerospace ground networks. To ensure the correctness of data transmission, the TCP data transmission protocol has been used typically. However, the standard TCP congestion control algorithm is incompatible with networks with a long time delay and a large bandwidth, resulting in low throughput and resource waste. This article compares recent studies on TCP-based acceleration algorithms and proposes an acceleration algorithm based on the learning of historical characteristics, such as end-to-end delay and its variation characteristics, the arrival interval of feedback packets (ACK) at the receiving end and its variation characteristics, the degree of data packet reversal and its variation characteristics, delay and jitter caused by the security equipment’s deep data inspection, and random packet loss caused by various factors. The proposed algorithm is evaluated and compared with the TCP congestion control algorithms under both laboratory and ground network conditions. Experimental results indicate that the proposed acceleration algorithm is efficient and can significantly increase throughput. Therefore, it has a promising application prospect in high-speed data transmission in aerospace-ground service networks.

## 1. Introduction

The distance between the sending and receiving ends of an aerospace-ground service network can exceed several thousand kilometers. Therefore, data transmission between the sending and receiving ends represents ultra-long-distance optical fiber transmission through a special line. It should be noted that without using a relay, the maximum effective transmission distance of an optical fiber is tens of kilometers, as optical signals attenuate to a certain extent to meet transmission bandwidth requirements. Accordingly, relay stations must be added to the transmission route to compensate for optical signal attenuation to realize ultra-long-distance transmission. However, the bit error rate (BER) increases with the number of used relay stations, which can result in packet loss and cause a packet error during data transmission. Namely, in ultra-long-distance optical fiber transmission over a special line, packet loss and an error in data transmission are caused by the attenuation of optical signals and BER, not by congestion on a physical link. However, in the standard TCP protocol, packet loss is treated as link congestion, thus reducing the transmission rate. Furthermore, this processing mechanism contradicts the reality of ultra-long-distance optical fiber transmission through a dedicated line, which results in bandwidth waste. The TCP protocol ensures data flow reliability using sequence confirmation and packet retransmission mechanisms. In addition, it achieves excellent adaptability under various network conditions and, thus, has significantly contributed to the rapid development and popularization of the Internet. However, the TCP protocol was designed more than two decades ago; consequently, it is unsuitable to model high-bandwidth, long-delay services in current ground networks. When packets are lost or delayed along the network path, the throughput of a TCP connection is significantly reduced. As a result, bandwidth is frequently underutilized, causing idle and unexploited bandwidth. Therefore, using the TCP will significantly increase long-distance data transmission and slow application response time, and it can even cause failure in data transmission. The literature [1,2] proposed some quantum logic gates and proved the success of the operations in implementing these gates. The literature [3,4,5,6] proposed a multi-qubit system consisting of two trapped ions coupled in a laser field. These devices may provide the next-generation design for quantum computers. To adapt to the current network characteristics of wide bandwidth and long delay, it is necessary to modify the TCP design to increase the transmission rate.

This article discusses the application of lightweight learning-based congestion control. The term “lightweight” refers to a type of congestion control algorithm that does not include deep learning, such as heuristic algorithms, utility functions, or gradient descent. A lightweight algorithm requires short training time and has a low cost, which makes it “light.” In addition, it accelerates TCP transmission and improves TCP connection stability by improving the standard TCP protocol and its handling of congestion, and the algorithm can detect and compensate for packet loss accurately and in a timely manner.

## 2. Materials and Methods

The TCP protocol was developed based on the RFC793 standard document published by the Internet Engineering Task Force (IETF) in 1981. The early development of the TCP protocol considered the effects of a transmission environment on the transmission rate, and both sender and receiver employed a sliding window strategy to control the data flow dynamically. However, as network services became more complex, it has been found that a simple flow control considers only the receiver’s accepting capacity. Nevertheless, from a macro perspective, the entire network contains a large number of routers and other network devices, and their storage and forwarding functions can affect the network’s congestion. Still, relying only on the receiver’s information cannot mitigate the effect of congestion by other network devices. This shortcoming caused a TCP collapse in 1986, resulting in a reduced link throughput between the LBL and UC Berkeley from 32 kbps to 40 kbps. Since then, researchers have recognized the critical nature of congestion control protocols, and pertinent research results have rapidly emerged [7,8].

The first congestion control algorithm was proposed by Van Jacobson et al. [9,10], which introduced mechanisms, such as slow start and congestion avoidance, for the first time. However, this type of algorithm immediately executes the slow-start strategy after judging the link as congested. This is because the link frequently reduces the size of the windows sent, impacting bandwidth utilization. In [11,12], a TCP-Reno algorithm was proposed to solve the bandwidth utilization problem by adding a fast recovery mechanism based on TCP Tahoe. Since the Reno algorithm can ensure network stability but not optimal resource utilization [13], in [14], a BIC algorithm, which consists of the binary searching and linear growth stages, was proposed. The Reno algorithm was modified in [15], and a modified TCP-Reno algorithm was developed. Further, in [16], the TCP-BIC algorithm was enhanced, and the TCP cubic algorithm, which improves the TCP-BIC algorithm’s window adjustment method, was developed. The TCP cubic algorithm is a default congestion control algorithm of the current Linux and Android kernels.

The conventional TCP protocol is incapable of correctly distinguishing the causes of packet loss, and it performs only indiscriminate window-reduction operations, thus limiting future network transmission efficiency enhancements [17,18,19]. In [20], a performance-based congestion control (PCC) protocol and a rate control mechanism were proposed to address the two mentioned issues. The proposed algorithm could increase the network’s transmission bandwidth, but the convergence rate was extremely slow. However, research on the PCC algorithm provided a large amount of information for subsequent analyses. Further, Google proposed an innovative congestion control scheme in 2016 named the dubbed BBR (Bottleneck Bandwidth and Round-trip propagation time) [21,22]. Certain concepts in the BBR are consistent with the PCC algorithm. Nonetheless, in [21,22], it was demonstrated that the BBR algorithm performed excellently in environments with a high bandwidth, long delay, and high packet loss rate.

Unlike traditional congestion control algorithms, the BBR algorithm uses the bandwidth-delay product (BDP) [12] as an identification indicator rather than the packet loss or long transmission delay to identify network congestion. When the total number of data packets in the network exceeds the BDP value, the BBR algorithm considers the network congested. Therefore, the BBR algorithm can be referred to as a congestion-based control algorithm. It should be noted that it is impossible for network data flow to achieve both an enormous link bandwidth and a very small network delay simultaneously. Accordingly, the BBR algorithm detects network capacity regularly, measures maximum link bandwidth and minimum network delay alternately, and then uses their product to determine the congestion window size. The congestion window can be used to characterize network capacity, providing a more accurate identification of congestion. Because of the BBR algorithm’s unique mechanism for measuring congestion window size, it neither increases the number of congestion windows indefinitely like ordinary congestion control algorithms nor uses the buffer of the switch node, thus avoiding the emergence of buffer bloat (buffer overflow) [13], which shortens the transmission delay significantly. Another advantage of the BBR algorithm is that it measures network capacity actively, adjusting the congestion window. In addition, the autonomous adjustment mechanism enables the BBR algorithm to control the data flow sending rate independently. In contrast, ordinary congestion control algorithms only calculate the congestion window, whereas the TCP protocol completely determines the data flow sending rate. As a result, when the data flow sending rate is close to the link’s bottleneck bandwidth, there is data packet queuing or data packet loss due to the rapid increase in the sending rate.

Initially, the BBR drew great attention from researchers and was considered a paradigm-shifting achievement in the field of congestion control. However, with research progress, it has been discovered that the BBR protocol has several shortcomings, including a slow convergence speed in the bandwidth detection stage, a low sensitivity, and a lack of consideration for delay and jitter.

## 3. Transmission Acceleration Using BoostTCP

This paper proposes the BoostTCP, which represents a learning-based TCP transmission acceleration method based on transmission history learning. By improving the judgment and handling of congestion, the BoostTCP can judge and recover packet loss more accurately and rapidly, thus accelerating TCP transmission and increasing TCP connection stability.

### 3.1. Improved Congestion Judging and Handling Mechanism

Many congestion estimation and recovery strategies were developed for standard TCP over the last two decades to meet network requirements under different conditions. The fundamental premise was that a packet loss represented a result of congestion. However, this assumption does not hold for a transmission network with ultra-long-distance special-line optical fiber. In such a network, packet loss is typically caused by the BER of long-distance transmission, not by congestion-related factors. Therefore, standard TCP can frequently enter an excessively conservative transmission state. Meanwhile, when a network path contains deep-queue network devices, packet loss does not occur for a long period after the congestion occurs. The standard TCP is insensitive to congestion, resulting in excessive transmission, which not only affects network congestion but can also cause significant packet losses. As a result, the TCP enters a lengthy recovery phase for packet loss, resulting in transmission stagnation. All of these factors contribute to the poor performance of the standard TCP protocol for an ultra-long-distance optical fiber transmission network with a special line.

Considering both packet loss and delay variation, the proposed BoostTCP algorithm can dynamically learn the network path characteristics of each specific connection during data transmission, including end-to-end delay and its variation characteristics, the arrival interval of feedback packets (ACK) at the receiving end and its variation characteristics, the degree of data packet reversal and its variation characteristics, delay and jitter caused by the security equipment’s deep data inspection, and random packet loss caused by various factors. These characteristics are monitored in real time and analyzed holistically to derive precursor signals and available bandwidth that reflect congestion and packet loss along the TCP connection network path. They also determine the degree of congestion and the transmission rate, and show whether the congestion recovery mechanism is compatible with the available bandwidth on the current path and can achieve accurate and timely packet loss judgment and recovery.

Based on network characteristics, the congestion degree and available bandwidth can be estimated accurately and in a timely manner. When congestion occurs, the transmission is realized based on the mentioned result. The unnecessarily slow data transmission rates caused by BER-induced packets can be avoided in an ultra-long-distance optical fiber transmission network with a special line. Specifically, the advanced congestion judgment and control algorithm of BoostTCP mainly uses the two following mechanisms: Prevent excessive conservative transmission and Prevent congestion deterioration.

#### 3.1.1. Prevent Excessive Conservative Transmission

Because the current TCP protocol stack has difficulty determining the cause of a packet loss (caused by network congestion) and the actual bandwidth available on the connection path following the packet loss, restoration has been typically performed to reduce the transmission rate significantly. This mechanism results in an idle path bandwidth, which is one of the primary reasons for TCP’s inefficient transmission performance.

The ultra-long-distance optical fiber transmission network with a special line often has sufficient bandwidth but a relatively long delay. In such a case, both the initial sending window and the current TCP protocol stack’s sending window increase at a relatively conservative rate. BoostTCP begins sending data with a large initial sending window and rapidly increases the sending window size to reach the upper limit of available bandwidth in the shortest time.

In an ultra-long-distance optical fiber transmission network with a special line, the BoostTCP congestion judge algorithm considers network characteristics, determining whether packet loss results from network congestion or not. As packet loss occurs as a result of random errors in an optical fiber network, and the transmission rate increases and is maintained at a higher rate, the rate is adjusted instantly when real congestion occurs. Namely, the bandwidth closest to the available bandwidth on the current path is used to perform transmission, and a slightly lower transmission rate is used to clear the queue on the path, which contributes to the recovery of nodes in a congested network. Moreover, transmission behavior is maintained to be consistent with and related to the network state. The judge algorithm eliminates idle bandwidth, resulting in a faster, more consistent transmission rate.

#### 3.1.2. Prevent Congestion Deterioration

Congestion may also occur in an ultra-long-distance special-line optical fiber transmission network due to a large number of networks and relays. In addition, congestion may worsen if not handled properly, causing two problems. First, the time required for retransmission and hole filling will be extremely long due to the high packet loss rate, and as a result, the TCP transmission window will become stuck for an extended period, and transmission will become slower or even fail. Second, retransmission is required due to increased packet loss; the retransmission rate increases while the effective data rate declines. Therefore, users will notice that although online traffic increases, the actual application rate does not change.

BoostTCP determines the congestion degree in real time and slows it down to prevent congestion from deterioration and reduce the number of lost packets, resulting in faster and smoother transmissions with an effective data rate.

In summary, the BoostTCP congestion judgment algorithm is an automatic state machine that considers various network characteristics along the transmission path. Its function is to learn and improve congestion judgment skills intelligently in a connection-by-connection manner. Since learning the network characteristics requires data accumulation, and the BoostTCP algorithm allocates resources to each TCP connection, the optimal application scenario for the BoostTCP algorithm is a long-connection scenario rather than a high-concurrency scenario, which is satellite payload data in this article.

### 3.2. Fast Prediction-Based Packet Loss Judgment and Recovery Mechanism

The standard TCP protocol stack determines packet loss in two ways, based on the number of Dup-ACKs received at the receiving end and based on the ACK timeout. When a large number of packets are lost, the ACK timeout has been frequently used to determine the timeout condition and initiate a retransmission. It should be noted that packet loss is frequently sporadic in modern networks, and it is not uncommon for multiple data packets to be lost concurrently on a connection. As a result, the standard TCP protocol frequently relies on timeouts to retransmit data to fill gaps, resulting in a waiting state of several seconds, which can even last up to ten seconds. As a result, the transmission may pause for an extended period or even disconnect entirely, which can affect the standard TCP efficiency significantly.

In addition to the two methods used by standard TCP, the BoostTCP’s packet loss judgment mechanism uses a dynamic self-learning algorithm to predict packet loss based on the network characteristics of the TCP connection path. The prediction algorithm considers network characteristic factors similar to those considered by the self-learning algorithm for BoostTCP congestion detection. The BoostTCP packet loss detection algorithm calculates a probability of loss for each packet sent but not confirmed by the other party’s ACK. The probability changes as the transmission process continues. When the probability reaches a certain value, the algorithm considers the data packet lost and initiates retransmission immediately. This mechanism significantly reduces the likelihood of the TCP transmission relying on timeout and determining the packet loss, allowing it to fill holes faster, transmit data more smoothly, and achieve a higher average transmission rate. This packet loss-to-retransmission mechanism, which is faster than the standard TCP, is beneficial for maintaining faster and smoother data transmission in ultra-long-distance special-line optical fiber transmission networks.

Due to the untimely packet loss detection of standard TCP, its transmission efficiency is frequently very low, and transmission quality is unstable, which is difficult to predict and impacts user experience. BoostTCP acceleration can predict packet loss in real time and recover the lost packets on time. The transmission is smoother and faster, significantly improving the user experience.

### 3.3. Congestion Control Algorithm

The flowchart of the BoostTCP congestion control algorithm is presented in Figure 1. It defines the smoothed throughput rate, which can reflect the actual throughput rate and roundtrip time, and controls the growth mode of the congestion window (CWND) based on different factors, such as the actual throughput rate and roundtrip time. The smoothed throughput rate variation is used to determine the most suitable CWND growth mode, which follows the principle of maximum throughput rate. As long as increasing the CWND value improves the smoothed throughput rate, the CWND value will be continuously increased. However, BoostTCP does not use the smoothed throughput rate to determine the necessary CWND reduction, and the CWND value to reduce is determined based on packet loss.

CWND growth can be classified into three types: exponential growth, linear growth, and termination. The exponential growth mode assumes that the current CWND value is one. After the first, second, and third increases, the CWND value is two, four, and eight, and further changes follow the exponential trend. Each time the CWND value increases linearly, it is increased by a fixed value. It should be noted that the CWND value does not increase during the termination stage and remains constant.

The specific steps of the BoostTCP congestion control algorithm are as follows:

**Step 1**. In the initial state, set the smoothed throughput rate to B = 0 and the growth mode (GM) of CWND to exponential growth;

**Step 2**. Every time a new data package is sent, record the sending time T_S_ of the package and the total amount of data F_S_ that has been sent and has not been acknowledged (ACKed) yet;

**Step 3**. When the ACK response is received, if the ACK corresponds to one or more data packages that have been sent and there are no retransmitted data packages, the data package in the acknowledged messages with the highest sequence number (SEQ) is selected, and the following parameters called instant throughput rate and smoothed throughput rate are calculated:

The instant throughput rate is calculated according to B_C_ = F_S_/(T − T_S_), where T denotes the current time; T_S_ denotes the sending time of the data package with the highest SEQ; and F_S_ denotes the total data amount that has been sent at the time T_S_ but has not been subject to ACK yet. As mentioned above, T_S_ and F_S_ are recorded when the data package with the highest SEQ is sent.

The smoothed throughput rate is obtained by B = (1 − α)B’ +αB_C_, where α denotes a constant parameter and B’ denotes the previous smoothed throughput rate set in the initial state or obtained in the previous calculation iteration. BoostTCP uses a first-order exponential smoothing formula to compute the smoothed throughput rate. This is because network delay often fluctuates constantly due to various reasons, causing the real-time throughput rate to fluctuate accordingly. After smoothing, some high-frequency noise can be eliminated, and the network throughput can be estimated more accurately;

**Step 4**. Determine the variation state of the smoothed throughput rate B and control the CWND growth mode GM accordingly. Particularly, the two following situations are possible:○If B is higher than the previous smoothed throughput rate set in the initial state or obtained in the last calculation and exceeds the set value γ, then the GM is an exponential GM;○If B decreases three times in a row and the total amount of the three reductions is not less than the preset value of Δ, then judge the SRTT value: if SRTT ≤ η · RTTMIN, then the GM is a linear GM; otherwise, the GM is a termination GM. SRTT denotes the smooth roundtrip time, and RTTMIN denotes the minimum roundtrip time.

**Step 5**. If packet loss occurs at any time, set CWND = β · CWND and the GM to a termination GM when entering the recovery mode.

**Step 6**. Set GM to the exponential GM when exiting the recovery state as being recovered from the congestion state and perform operations similar to the above initial state. After exiting from the recovery mode, the smoothed throughput rate B is not cleared but performs operations similar to the initial state based on the original smoothed throughput rate.

The definitions and values of the above α, β, γ and other parameters are shown in Appendix A.

### 3.4. Implementation Architecture

The BoostTCP consists of several modules, as shown in Figure 2. The modules are explained in the following:○Learning state machine: This is an information and control hub of BoostTCP, which accumulates knowledge about network paths and makes real-time decisions about the transmission of specific connections, such as the rate at which data are transmitted and the timing of data retransmission;○Traffic monitor: This module extracts and learns the external features of each TCP flow and records and maintains the learning state machine;○Packet loss monitor: This module monitors packet loss and determines the most probable cause of data loss using a learning state machine, for instance, whether the loss is caused by simple random packet drops or network congestion;○Congestion controller: This controller executes the core congestion control logic based on a learning state machine;○Exception handler: This module leverages knowledge of the learning state machines to identify flaws in peer TCP stacks or certain devices along the data transmission path, such as security detection devices. This module is used to detect specific characteristics of TCP to ensure maximum acceleration. Exception handlers also contribute to the knowledge accumulation of learning state machines;○Window controller: This controller calculates the size of the TCP broadcast window and balances incoming packets from the LAN and WAN sides;○Resource manager: This module tracks and controls system resources, including memory and computing power, and dynamically balances system resource consumption across all active TCP flows. The knowledge of learning state machines is the input to resource management.

### 3.5. Deployment Location

As an acceleration engine, BoostTCP follows the network driver interface specification and is located between the protocol stack and the hardware network interface card (NIC). It is fully compatible with the standard TCP protocol and does not attempt to replace the original TCP protocol stack in the operating system. When an application continues to interact with the TCP stack of the operating system in which it resides, BoostTCP is completely transparent to the application. When traffic is routed through the BoostTCP module, BoostTCP accelerates it by changing the timing of data packet transmission and retransmission without changing the data content or TCP encapsulation format. The position of BoostTCP in a multi-layer network architecture is presented in Figure 3.

## 4. Experimental Results

### 4.1. Experimental Environment

The study simulated the network environment with the TC and used the Reno, BBR, BoostTCP, and standard TCP algorithm in the simulation tests. The performance indicators of throughput, fairness, and preemption were compared for scenarios with varying bandwidth, delay, and random packet loss rate. The experiment was conducted on five Inspur NF5280M5 servers equipped with two Intel Xeon-GoXD 6136 (3.0 GHz/12-core) processors and 64 GB memory. The simulation network structure is presented in Figure 4. In the presented structure, Server 1 acted as a sender, employing the Cubic and Reno congestion control algorithms; Server 2 acted as a sender, employing the BoostTCP algorithm; Server 3 acted as a sender, employing the BBR algorithm; Server 4 acted as a receiver; and, lastly, Server 5 acted as a simulated controller for the designed network environment. The two network ports on Server 5 were connected to the switch and Server 4, forming a network bridge. TC managed the delay and packet loss and simulated the environment of a wide-area network.

The experimental topology was an end-to-end configuration with 1 Gbps network bandwidth. Multi data flows were sent from the sender to the receiver at the specified network delay and packet loss rate, with a default data packet size of 500 MB.

Four well-known congestion control methods were used in the experiment: three lightweight learning-based methods (Reno, BBR, and BoostTCP) and the conventional Cubic algorithm as a contrast method. To simulate the characteristics of a long-distance, long-delay, and low-packet-loss-rate data transmission in a real aerospace business network, the network delay in the experiment was set to 5 ms, 10 ms, 20 ms, 30 ms, 40 ms, 50 ms, 80 ms, and 100 ms, and five different random packet loss rates were used: zero, 0.01%, 0.05%, 0.1%, and 0.5%.

### 4.2. Results Analysis

To analyze the performance of the proposed method, it was tested and compared with the other algorithms regarding throughput, fairness, and preemptibility.

#### 4.2.1. Average Throughput

The average throughput curves of the four congestion control algorithms at a bandwidth of 1 Gbps and network delays of 20 ms, 30 ms, 50 ms, and 80 ms are presented in Figure 5, Figure 6, Figure 7 and Figure 8, respectively. The throughput was tested three times and averaged at each packet loss rate. The average throughput rate of BoostTCP was always the highest among all algorithms, and its predominant position became more apparent with packet loss, achieving a 26–41% enhancement over the BBR. The average throughput rate of the Reno algorithm was always the lowest among all algorithms. The average throughput rate of the Cubic algorithm decreased the most rapidly with the packet loss rate among all algorithms.

The average throughput results of the four congestion control algorithms at a packet loss rate of 0.05% and a bandwidth of 1 Gbps is shown in Figure 9. The results presented in Figure 9 were averaged for each time delay. The average throughput rate of the BoostTCP algorithm was always the highest among all algorithms, and its advantage over the other algorithms became even more obvious at a longer time delay. Compared to the BBR algorithm, the average throughput of the BoostTCP algorithm was nearly identical in the early stages and increased to 2.3 times that of the BBR algorithm at a network delay of 100 ms. Among all algorithms, the Reno algorithm had the lowest average throughput. The average throughput rates of the Cubic and BBR algorithms decreased more rapidly than that of the BoostTCP algorithm with a time delay.

According to the experimental results, BoostTCP had the highest average throughput under most conditions among all algorithms. This was because BoostTCP’s bandwidth detection mechanism was based on learning transmission history and considered actual throughput and roundtrip time factors, which could fully use the link’s excess bandwidth. Compared to the BoostTCP algorithm, the BBR algorithm’s throughput was less, and the rate dropped more rapidly with a longer time delay. The reasons for this were that the convergence speed of the BBR algorithm was too slow, the sensitivity of the bandwidth detection stage was insufficient, and issues, such as delay and jitter, were ignored. The Reno algorithm had the lowest average throughput for various random packet loss rates among all algorithms. Further, the Cubic algorithm had a higher throughput than the Reno algorithm, but the throughput rapidly decreased as the rate of random packet loss increased. Since the Cubic congestion control was based on packet loss, this was necessary. Random packet loss could significantly impact its judgment of network conditions, resulting in performance degradation.

The main idea of the BBR algorithm is to detect the maximum bandwidth and minimum roundtrip time continuously and alternately and then estimate overall network congestion using the two extreme values. Thus, the minimum roundtrip time accuracy is critical in determining the BBR algorithm’s impact on network congestion. The ground network environment’s primary characteristics are a high delay and sufficient bandwidth. In this case, the minimum roundtrip time is no longer capable of responding to network congestion accurately. Therefore, if the BBR algorithm continues to estimate the congestion window using the detected minimum roundtrip time, the estimated CWND value of the congestion window will be less than the link’s actual ideal capacity. Further, reduced CWND limits the sender’s sending rate, causing the bandwidth value measured by the BBR algorithm in detecting the link’s maximum bandwidth to be less than the link’s best achievable bandwidth. For instance, a lower maximum bandwidth results in a lower CWND value. As a result, the BBR algorithm can operate only at a reduced rate, thus causing significant network resource waste.

BoostTCP dynamically learns the network path characteristics of each TCP connection during transmissions, such as the end-to-end delay and its variation characteristics, the arrival interval and variation characteristics of the receiver’s feedback data packet (ACK), the degree of data packet reversal and its variation characteristics, delay and jitter caused by deep data inspection by security equipment, and random packet loss caused by various factors. While tracking these characteristics in real time, BoostTCP analyzes them holistically and derives precursor signals that reflect congestion and packet loss along the TCP connection network path. Further, it determines the congestion degree based on the results of the dynamic, intelligent learning processes; determines the transmission rate and the congestion recovery mechanism that are compatible with the available bandwidth on the current path; and then performs the packet loss judgment and recovery accurately and in a timely manner. The BoostTCP algorithm can detect congestion in real time, automatically slow down, avoid mechanism congestion caused by excessively aggressive transmission, and accurately identify packet loss caused by random error codes. Thus, high-speed transmission is maintained and transmission behavior is smoother, which indirectly increases the effective data transmission rate.

#### 4.2.2. The Fairness of Single Algorithm for Multiple Flows

To investigate the fairness of sharing link bandwidth when multiple flows coexist in the same scheme, this study sent one data flow at 0 s, 10 s, and 20 s in the test to determine whether three data flows can finally share the link bandwidth evenly, as well as the time required to evenly share bandwidth and reach convergence. Using a 1 Gbps link bandwidth, 100 ms network delay, and zero random packet loss rate as an example, the tested fairness of each scheme is summarized as follows.

The fairness results of the algorithms are presented in Figure 10, where it can be seen that the BBR had a higher throughput rate than the other algorithms when only one data flow was used. However, when two data flows of 10 s and 20 s are added, the throughput rates of the two data flows significantly differed. After 30 s, the throughput rates of the three data flows fluctuated, indicating that the BBR algorithm was unable to achieve an effective link bandwidth share. The results of the Cubic congestion algorithm for a network delay of 100 ms and a packet loss rate of zero are presented in Figure 11. After adding data flows at 10 s and 20 s intervals, the Cubic algorithm could average the throughput of three flows and ensure efficient link bandwidth sharing. The results of the BoostTCP congestion algorithm at a network delay of 100 ms and a packet loss rate of zero are presented in Figure 12. As demonstrated in Figure 12, the BoostTCP algorithm could maintain fairness between three data flows in a steady-state. However, the BoostTCP algorithm had a larger bandwidth-sharing fluctuation range than the Cubic algorithm, achieving an average value of 6.6 Mbps. BoostTCP could achieve a transmission rate of 320 Mbps after stabilization, which was faster than those of BBR and Cubic, indicating a more efficient use of network resources.

#### 4.2.3. Analysis Results of Preemption Ability

Different TCP connections have different bandwidth preemption levels in a real-world transmission network because they use different congestion control protocols. The bandwidth preemption level shows the ability to preempt bandwidth in terms of transmission performance. The greater the preemption capability is, the more efficiently network resources are used. The preemptive results of the BBR, Cubic, and BoostTCP algorithms are presented in the following figures.

Figure 13 and Figure 14 illustrate the preemptive test curves of the congestion algorithms for the zero network delay and packet loss rate. The first test was conducted with the Cubic algorithm, followed by the BBR algorithm 10 s later. At the moment, the Cubic and BBR algorithms coexisted, and the Cubic algorithm severely preempted the BBR’s bandwidth, resulting in no significant increase in the BBR algorithm’s throughput. BoostTCP was restarted after 20 s, after which congestion occurred. The BoostTCP algorithm’s throughput rate reached a stable value quickly, within 3 s, while the BBR algorithm’s throughput rate gradually increased. After 35 s, the three algorithms’ throughput rates converged to a steady-state. The BoostTCP algorithm had a higher throughput rate than the Cubic and BBR algorithms. The second test started with the BBR algorithm, and was followed by the Cubic algorithm 10 s later. Thus, the Cubic and BBR algorithms coexisted 10 s after the test began. The Cubic algorithm severely restricted the BBR’s bandwidth, resulting in a throughput rate of nearly zero. After 20 s, the BoostTCP algorithm was invoked and congestion occurred. The BoostTCP algorithm’s throughput rate reached a stable value quickly, within 3 s, while the BBR algorithm’s throughput rate gradually increased. After 35 s, the three algorithms’ throughput rates stabilized. The BoostTCP algorithm had a higher throughput rate than the Cubic and BBR algorithms.

The preemptive test curves of the congestion algorithm for a network delay of 80 ms and a packet loss rate of zero are shown in Figure 15. The third test began with the Cubic algorithm, and the BBR algorithm was run after a 10 s delay. After that moment, the Cubic and BBR algorithms coexisted, and their throughput rates were essentially identical. After 20 s, the BoostTCP algorithm was started. Congestion occurred during this period. The BoostTCP algorithm’s throughput rate rapidly stabilized after 3 s, whereas the BBR and Cubic algorithms’ throughput rates decreased slightly. Finally, the three algorithms’ throughput rates reached their steady states. The BoostTCP algorithm had a higher smoothed throughput rate than the Cubic and BBR algorithms.

The results of the three tests indicated that the BoostTCP algorithm had a better ability to preempt bandwidth than the BBR and Cubic algorithms under different delay conditions. Additionally, the results demonstrated that the BoostTCP algorithm was beneficial to the BBR and Cubic algorithms by assisting suppressed algorithms in resuming their normal throughput rates, demonstrating the BoostTCP algorithm’s correctness.

### 4.3. Test in Actual Environment

To validate the BoostTCP algorithm’s performance in real-world network transmission, a real-world network test was conducted analyzing the data transmission throughput rate between ground stations and satellite user centers. The real-world network test results of BoostTCP and standard TCP are presented in Figure 16, where the data transmission performances of the two algorithms were compared for a network consisting of seven ground stations and a satellite user center. The relationship between the increase in data transmission throughput rate and the network’s maximum bandwidth is depicted in Figure 17. The relationship between the speed-up ratio of data transmission throughput rate and network delay is shown in Figure 18.

As presented in Figure 16, the BoostTCP algorithm performed significantly better than the standard TCP algorithm. Data transmission rates between the satellite user center and seven ground stations were significantly increased. For instance, the speed-up ratio was typically tenfold and could reach seventyfold. The result indicates that the BoostTCP algorithm significantly increased data transmission throughput and effectively increased network resource utilization.

The relationship between the increase in data transmission throughput and the maximum network bandwidth is presented in Figure 17. As shown in Figure 17, when the network bandwidth increased from 300 MB to 2300 MB, the data transmission throughput rate increased significantly. The relationship between the speed-up ratio of the data transmission throughput rate and the network delay is displayed in Figure 18. When the network delay increased from 1 ms to 70 ms, the data transmission throughput rate’s speed-up ratio increased proportionately. As a result, the greater the network bandwidth and delay were, the greater the performance advantage of the BoostTCP algorithm was. The measured data have conclusively demonstrated that the BoostTCP algorithm is more suitable for networks with high bandwidth and a long delay than the conventional TCP algorithm.

## 5. Conclusions

Due to the high precision requirements for satellite payload data transmission via a ground network, the TCP protocol can be considered competitive. However, the limitations of the standard TCP protocol on bandwidth utilization for networks with a long delay and large bandwidth reduce data transmission efficiency. This article uses TC to design a WAN simulation environment. Four congestion control algorithms—Reno, BBR, BoostTCP, and Cubic—are tested and compared in terms of throughput, fairness, and preemptibility. The results indicate that BoostTCP is more adaptable to network conditions and has a significantly higher throughput than the other three algorithms. In addition, it is fairly distributed across multiple data flows and has relatively strong preemption capability when multiple protocols are used. Finally, the throughput of BoostTCP is verified and tested in a real-world environment, and the results indicate that the real-world performance is identical to that in a simulated environment. Therefore, the proposed TCP acceleration algorithm can be used to improve the performance of ground networks when transmitting satellite payload data. In recent years, as a new field of quantum technology, the implementation of a quantum algorithm and quantum network has received increasing attention from scholars. In the next step, we will discuss the application of quantum algorithms and quantum networks in aerospace-ground service networks.

## Figures and Tables

**Figure 1 sensors-22-09187-f001:**
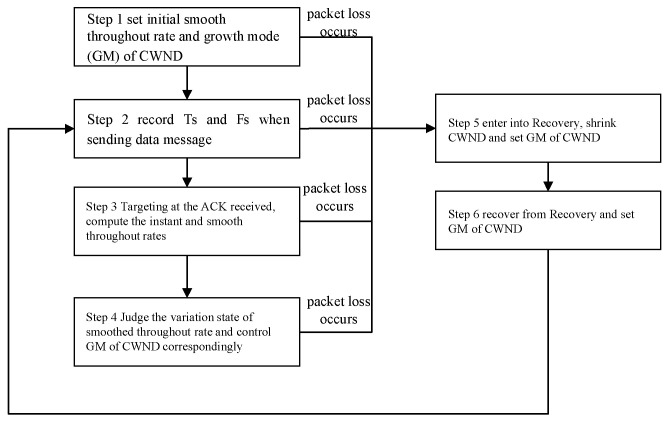
Flowchart of the BoostTCP congestion control algorithm.

**Figure 2 sensors-22-09187-f002:**
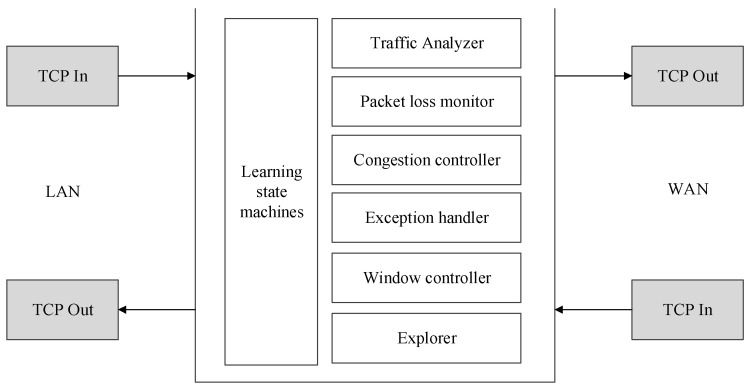
Schematic diagram of the BoostTCP modules.

**Figure 3 sensors-22-09187-f003:**
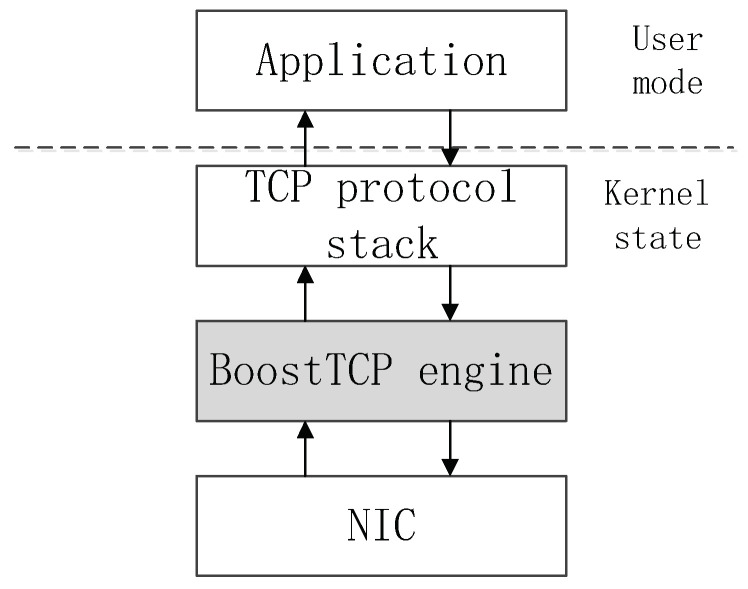
Illustration of the BoostTCP position in the multi-layer network transmission architecture.

**Figure 4 sensors-22-09187-f004:**
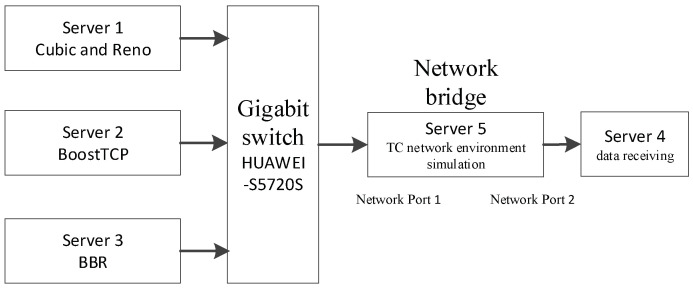
The experimental network structure.

**Figure 5 sensors-22-09187-f005:**
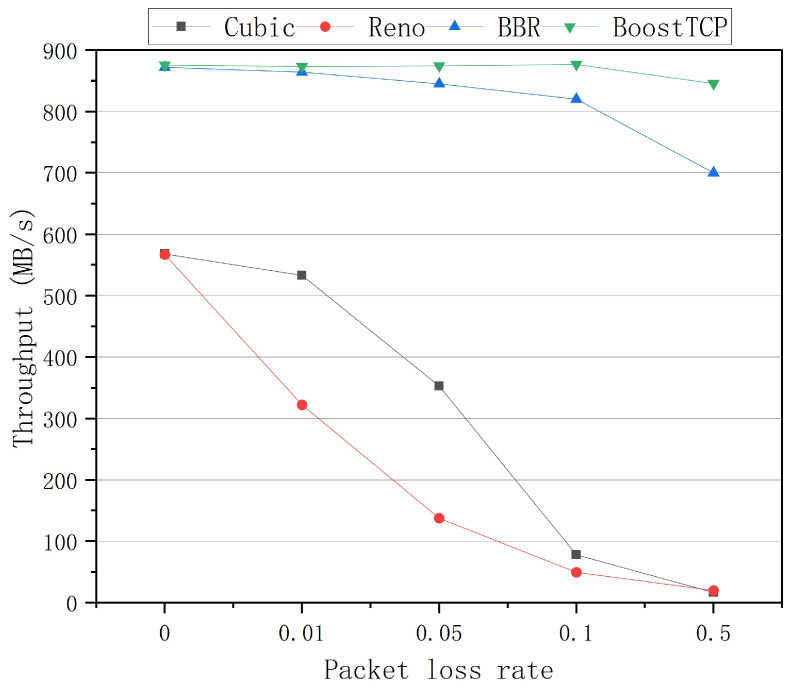
The throughput curves of the four congestion control algorithms as a function of the packet loss rate at a network delay of 20 ms.

**Figure 6 sensors-22-09187-f006:**
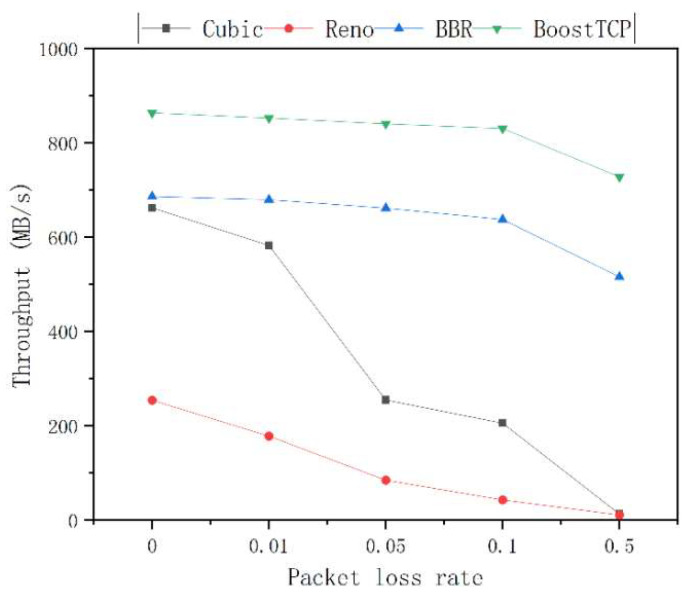
The throughput curves of the four congestion control algorithms as a function of the packet loss rate at a network delay of 30 ms.

**Figure 7 sensors-22-09187-f007:**
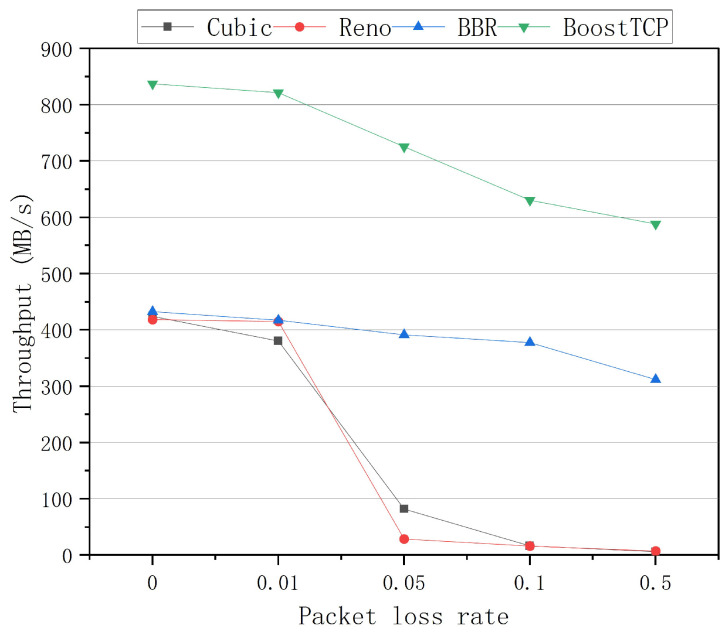
The throughput curves of the four congestion control algorithms as a function of the packet loss rate at a network delay of 50 ms.

**Figure 8 sensors-22-09187-f008:**
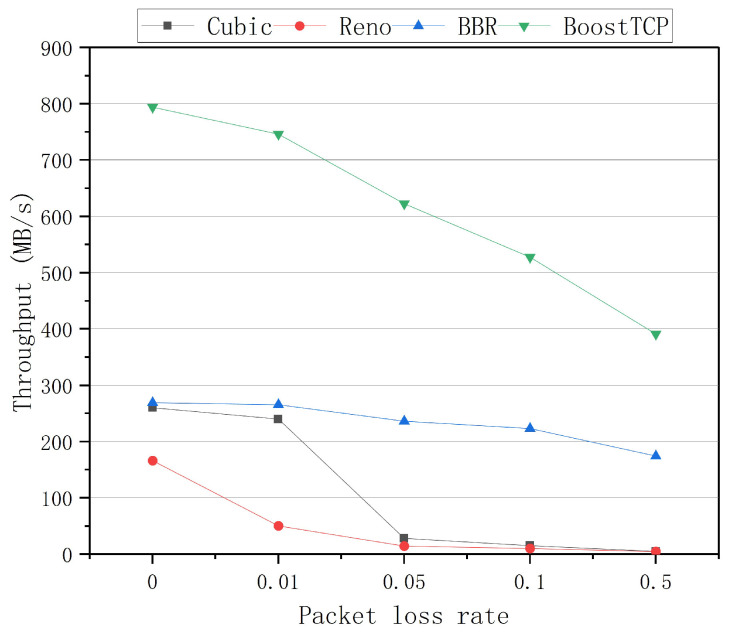
The throughput curves of the four congestion control algorithms as a function of the packet loss rate at a network delay of 80 ms.

**Figure 9 sensors-22-09187-f009:**
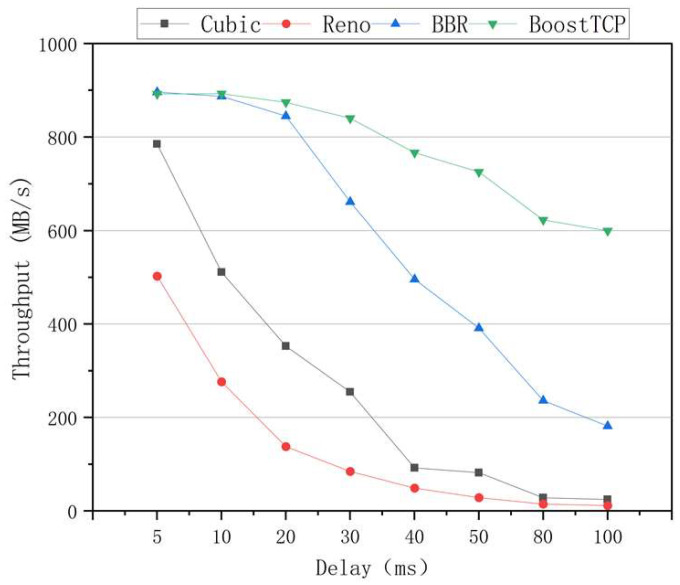
The throughput curves of the four congestion control algorithms versus the network delay at a packet loss rate of 0.05%.

**Figure 10 sensors-22-09187-f010:**
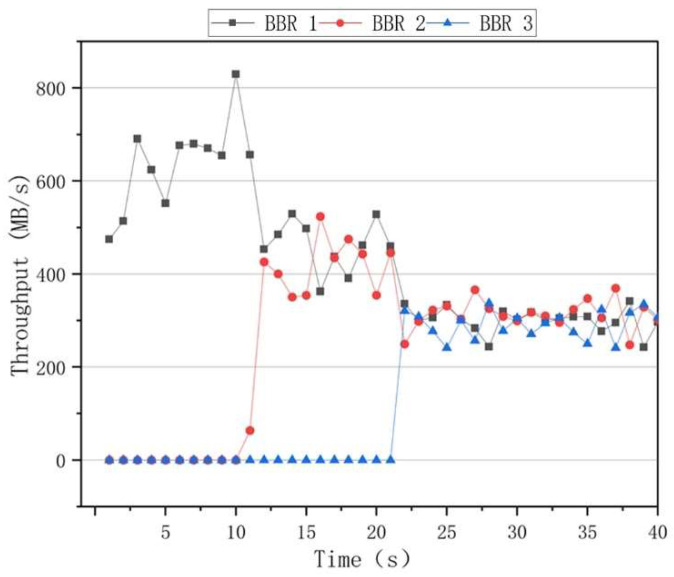
The fairness curves of the BBR congestion control algorithm at a network delay of 100 ms and a packet loss rate of zero.

**Figure 11 sensors-22-09187-f011:**
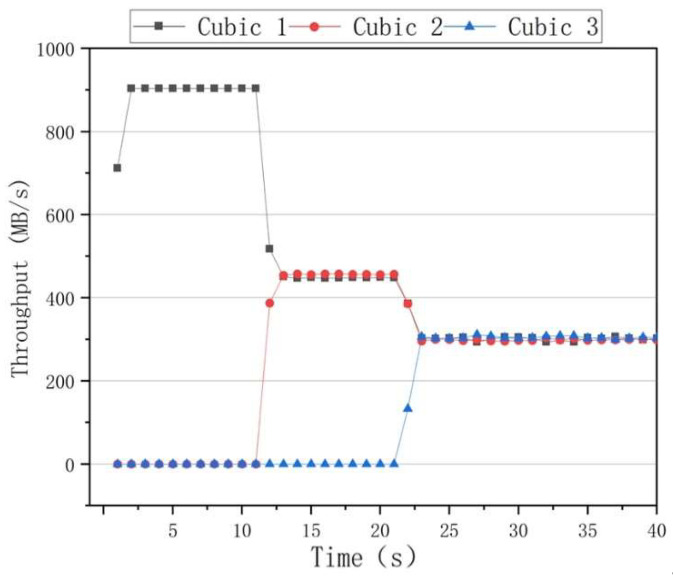
The fairness test curves of the Cubic congestion control algorithm at a network delay of 100 ms and a packet loss rate of zero.

**Figure 12 sensors-22-09187-f012:**
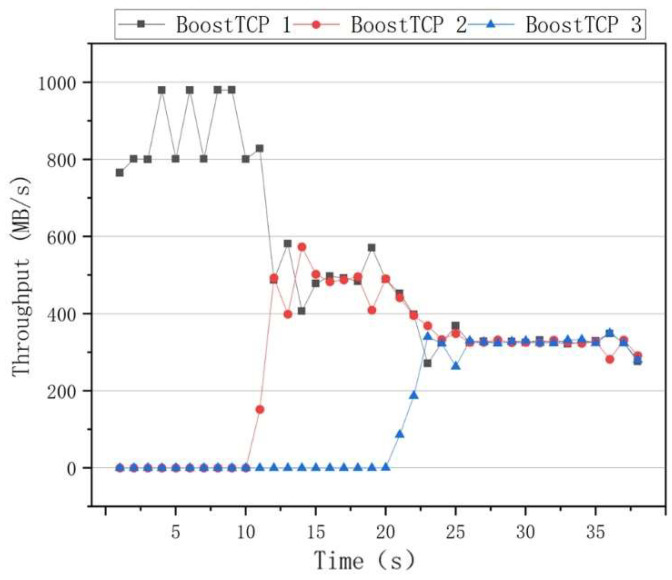
The fairness test curves of the BoostTCP congestion control algorithm at a network delay of 100 ms and a packet loss rate of zero.

**Figure 13 sensors-22-09187-f013:**
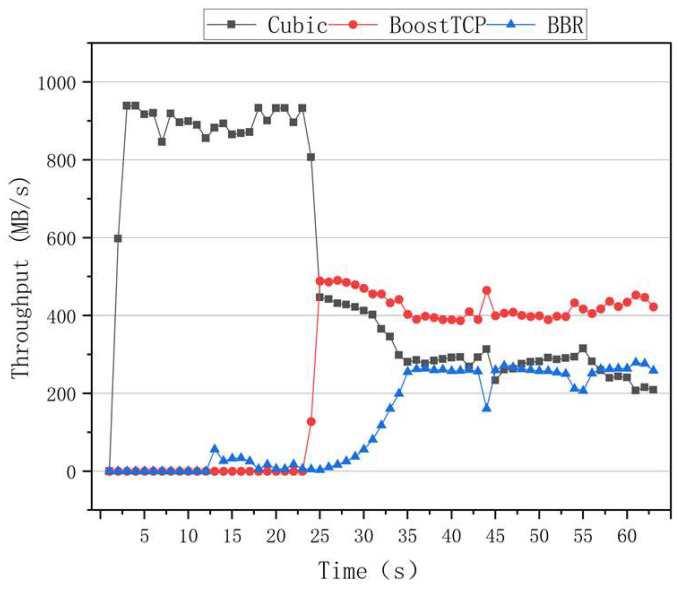
The preemptive test curves of the three congestion control algorithms at the zero network delay and packet loss rate.

**Figure 14 sensors-22-09187-f014:**
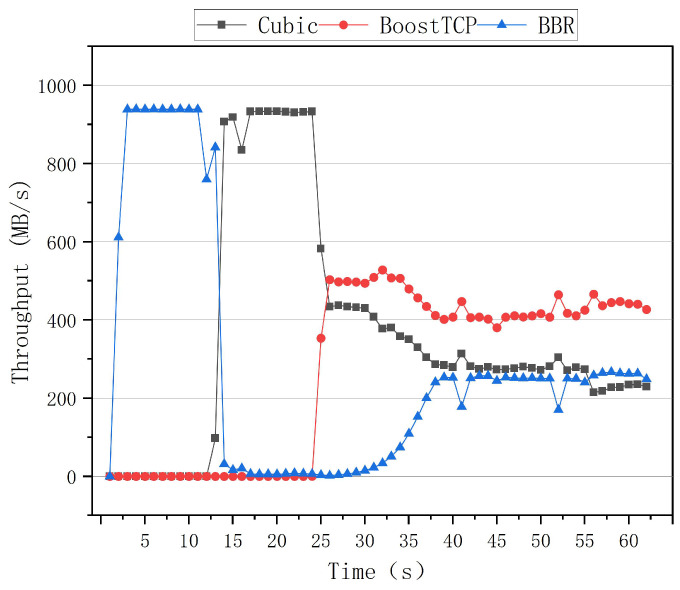
The preemptive test curves of the three congestion control algorithms at a network delay of zero and a packet loss rate of zero.

**Figure 15 sensors-22-09187-f015:**
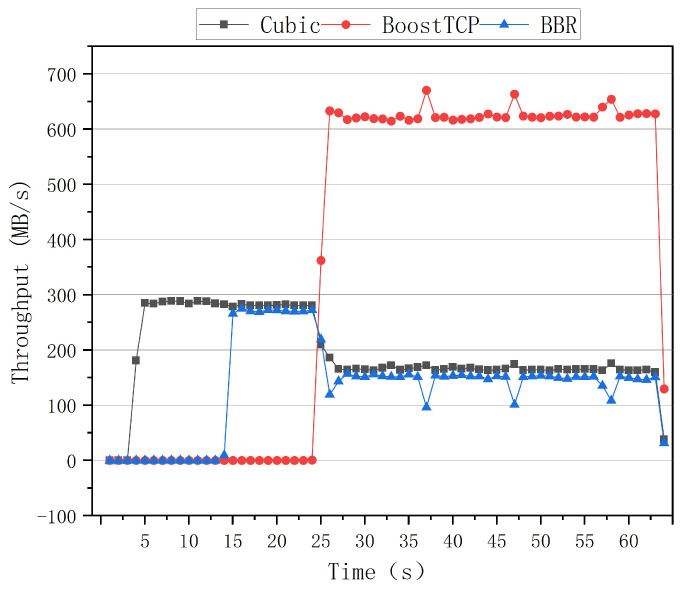
The preemptive test curves of the three congestion control algorithms at a network delay of 80 ms and a packet loss rate of zero.

**Figure 16 sensors-22-09187-f016:**
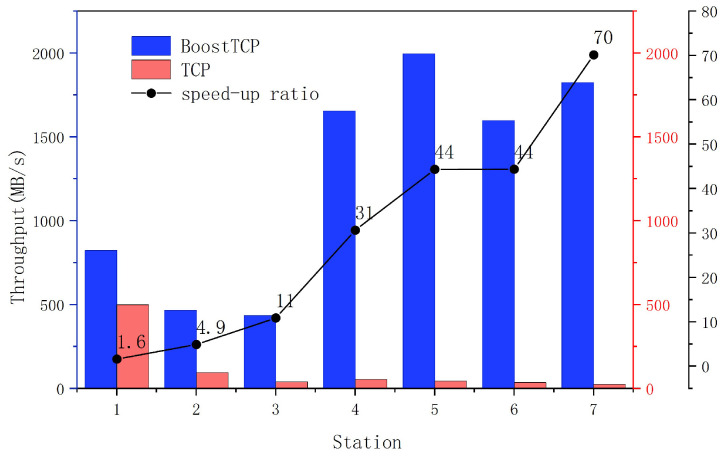
The actual network test results of the BoostTCP and ordinary TCP.

**Figure 17 sensors-22-09187-f017:**
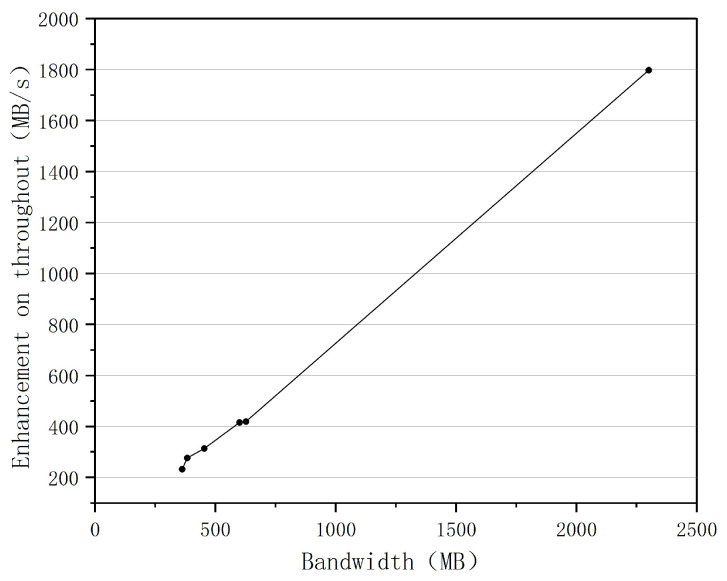
The throughout enhancement versus the maximum network bandwidth.

**Figure 18 sensors-22-09187-f018:**
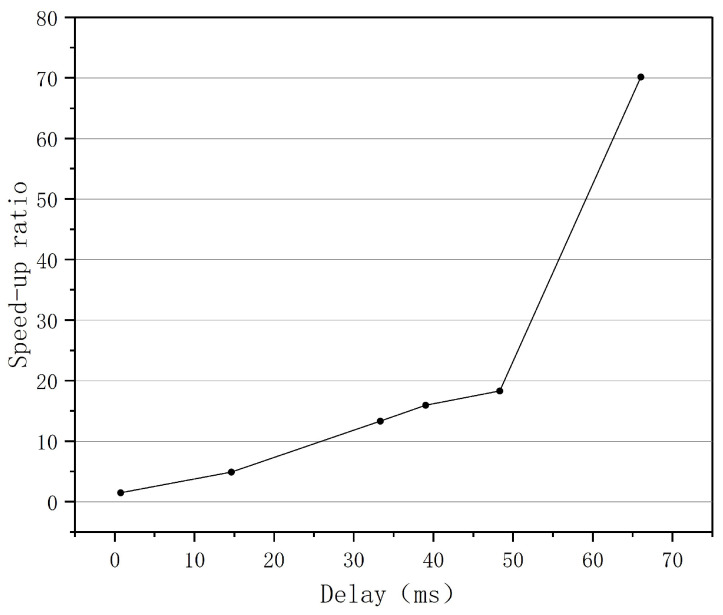
The speed-up ratio of data transmission versus the network delay.

## Data Availability

Not applicable.

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
