# Peer review of "A TCP Acceleration Algorithm for Aerospace-Ground Service Networks"

_sensors, 2022, doi:10.3390/s22239187_

Round 1

Reviewer 1 Report

Presented article describes a relevant problem for modern communication systems. Material is interesting and well illustrated. Theory has a backgroud in the form of computer simulations. Soundness of the article is scientific. A single point, which needs improvement, is English language of the text. Authors must reread it one more time, exclude grammar and other mistakes, change some word orders, etc. After corrections of the text, article can be recommended for publication.

Author Response

Thank you very much for your high comment on this paper. In order to improve the language, we invited native English-speaking editors to correct and verify the grammar, spelling, and punctuation in the paper. Hope our effort help audiences to better understand our research.

Reviewer 2 Report

Dear   Editor, 

Sensors

The subject of the transmission of satellite payload data is critical in the services provided by aerospace ground networks.  The data transmission protocol known as TCP is typically used to ensure the correctness of data transmission. They study a TCP acceleration algorithm for aerospace-ground service network. Therefore, the authors should extend the part about the   implementation of quantum algorithm and quantum network  in more general way as new fields interested by development of algorithms and  network. The   implementation of quantum algorithm and quantum network should be discussed for some real  physical systems in the introduction and cited:    J. Phys. A: Math. Theor.  45 (2012) 485305; Ann. Phys. 334 (2013) 47;J.Opt. Soc.Am 30 (2013) 1178;  Quantum Inf. Process, 13 (2014) 1947;PhysicsLettersA383(2019)1247-1254;Rep. Prog. Phys. 74 (2011) 104401.

  In conclusion,  I recommend the revised version for publication in   Sensors.

Author Response

Thank you for your high evaluation of this paper and providing the latest research trends in this field, which inspired us to revise the paper. We have updated the paper by discussing quantum algorithms as well as implementation of quantum network in the introduction part, and cited some documents of the field. In order to improve our algorithm, quantum and its related will be the next research direction.

Reviewer 3 Report

The proposal is clear and very interesting. I think it is important to consider the following elements in an updated version of this paper.

- what is the Beta value in the experiments and how does it impact the performance?

- more information on the learning machine would have been appreciated

- there are some fairness issues with BBR and CUBIC. This should be highlighted as an issue of the protocol.

- BBR is a response to delays in the buffers. What are the buffer sizes used during the experiments and does it impact the protocol performances ? Since the protocol is mainly exploiting loss as a congestion signal, large buffers may introduce large end-to-end delay.

- the proposal would not be applicable to QUIC where the idea of reusing the congestion control parameters of previous connections is currently discussion in the IETF working groups. it may be worth discussing these elements.

Round 2

Reviewer 3 Report

The authors answered my questions.

Some still remain (e..g impact of the buffer size on the fairness between CUBIC and BBR) but this should not prevent the publication of the paper.